# DISA tool: Discriminative and informative subspace assessment with categorical and numerical outcomes

**Leonardo Alexandre**[1,2,3]*, **Rafael S. Costa**[3,4], **Rui Henriques**[1,2]

**1** INESC-ID, Lisboa, Portugal, **2** Instituto Superior Técnico, Universidade de Lisboa, Lisboa, Portugal, **3** LAQV-REQUIMTE, Department of Chemistry, NOVA School of Science and Technology, Universidade NOVA de Lisboa, Caparica, Portugal, **4** IDMEC, Instituto Superior Técnico, Universidade de Lisboa, Lisbon, Portugal

* leonardoalexandre@tecnico.ulisboa.pt

**Data Availability Statement:** The first three datasets are available at UCI machine learning repository (https://archive-beta.ics.uci.edu/) with acess links: - https://archive-beta.ics.uci.edu/ml/

## Abstract

Pattern discovery and subspace clustering play a central role in the biological domain, supporting for instance putative regulatory module discovery from omics data for both descriptive and predictive ends. In the presence of target variables (e.g. phenotypes), regulatory patterns should further satisfy delineate discriminative power properties, well-established in the presence of categorical outcomes, yet largely disregarded for numerical outcomes, such as risk profiles and quantitative phenotypes. DISA (Discriminative and Informative Subspace Assessment), a Python software package, is proposed to evaluate patterns in the presence of numerical outcomes using well-established measures together with a novel principle able to statistically assess the correlation gain of the subspace against the overall space. Results confirm the possibility to soundly extend discriminative criteria towards numerical outcomes without the drawbacks well-associated with discretization procedures. Results from four case studies confirm the validity and relevance of the proposed methods, further unveiling critical directions for research on biotechnology and biomedicine. **Availability:** DISA is freely available at https://github.com/JupitersMight/DISA under the MIT license.

## Introduction

The discovery of discriminative patterns has proven essential to support predictive and descriptive tasks [1–6]. More specifically in gene expression data, discriminative patterns play an essential role to discover outcome-specific regulatory modules for knowledge acquisition, biomarking phenotypes of interest [7], or serve as the basis for drug targeting (e.g. cancer) after rigorous validation [8]. Discriminative pattern mining also plays a role in unraveling complex interactions in biological processes such as the condition-specific interplay among transcription factors in organisms [9]. In this context, patterns help mapping regulatory interactions, forming regulatory networks, that provide a vital information to better understand the evolution of the genes, as well as unique regulatory cascades elicited in response to stimuli, disease progression or drug action [10]. These discriminative properties towards an outcome of

datasets/echocardiogram- https://archive-beta.ics.uci.edu/ml/datasets/liver+disorders- https://archive-beta.ics.uci.edu/ml/datasets/breast+cancer+wisconsin+diagnostic The last dataset is made available by https://www.nature.com/articles/s41467-020-18008-4 The aforementioned sources are the original repositories, a secundary source is found in the following Github Repository:- https://github.com/JupitersMight/DISA/tree/main/Example.

**Funding:** This work was supported by the Associate Laboratory for Green Chemistry (LAQV), financed by national funds from FCT/MCTES (UIDB/50006/2020 and UIDP/50006/2020), INESC-ID plurianual (UIDB/50021/2020), the contract CEECIND/01399/2017 to RSC and the FCT individual PhD grant to LA (2021.07759.BD). This work was further supported by IPOscore with reference (DSAIPA/DS/0042/2018) and ILU (DSAIPA/DS/0111/2018).

**Competing interests:** The authors have declared that no competing interests exist.

interest can either be incorporated in the pattern discovery process [11, 12], or assessed after extracting classic informative patterns. In both cases, one or multiple interestingness measures, such as confidence [13], statistical significance [14] (probability of pattern occurrence against expectations) and/or discriminative power views [15, 16], are combined into pattern-centric models to aid medical decisions and study regulatory responses to events of interest [12, 17].

Although it is crucial to incorporate these discriminative criteria in the discovery task, existing contributions are generally focused on nominal outcomes [18, 19]. Nonetheless, many phenotypes of interest, such as molecular and physiological features, as well as risk scales or drug dosages, are quantitative variables in nature. In metabolic engineering the levels of production and/or degradation of certain organic compounds are continuous outcomes of interest [20, 21]. In such cases, to assess the ability of the underlying patterns to discriminate specific outcomes of interest, related work usually resorts to one of three following approaches: 1) distribution-based methods [22, 23], which explore properties of the distribution of continuous data, providing standard statistical measures on the distribution of the pattern-associated outcomes. In this context, Aumann and Lindell [23] consider measures such as the mean, with the possible alternatives of variance or median, to describe numerical distributions. An example of the aforementioned is an association rule like "*sex = female → mean wage = $7.90 p/hr*", where they guarantee the rule's discriminative properties by using classical measures, lift and confidence, and further ensure the validity of the outcome of interest by applying a Z-test; 2) discretization-based methods [24], which categorise the outcome variable in order to apply classic discriminative criteria in the discovery task. Well-known discretization methods categorise data based on frequency, user-inputted ranges, or more complex approaches such as the ones proposed by Alexandre *et al.* [25] where numerical variables are fitted and categorised according to a continuous distribution. While not the same as discretization, fuzzy-logic-based approaches can also be used in the presence of quantitative [26–28], and continuous variables [29], to extract informative patterns; and 3) optimization-based methods [30], which consider stochastic searches that follow the idea of natural selection and genetics (e.g., particle swarm optimization methods). Particles produced and modified along the evolution process are the targeted discriminative patterns, where both the pattern and the bounded range of relevant outcomes are optimized during the search [30]. While classic discriminative views are only prepared for nominal outcomes, these three classes of approaches are unable to establish an objective assessment of whether a given pattern is able or not to significantly discriminate a specific range of numerical outcomes.

To address these limitations, this work proposes a methodology to rigorously assess association rules with expressive patterns in the antecedent and numerical outcomes in the consequent, thus avoiding the discovery of spurious association rules (false positives). To this end, we introduce a novel distribution-based approach that inspects the differences between the distribution of a numerical outcome for all observations and a given pattern. To the best of our knowledge, there are no software packages able to assess association rules with robustness in the presence of numerical outcomes [31, 32]. Ergo, we propose DISA (Discriminative and Informative Subspace Analysis), a software package in Python to assess patterns with numerical outputs by statistically testing the correlation gain of the pattern against the overall data, identifying discriminative ranges of numerical outcomes tailored to each pattern.

## Background

Multivariate data can be structured in the form of a matrix $A = (X, Y)$, with a set of observations $X = \{x_1, \ldots, x_N\}$, variables $Y = \{y_1, \ldots, y_M\}$, and elements $a_{ij} \in \mathbb{R}$ observed for observation $x_i$ and variable $y_j$. One way to extract patterns from this data structure is through the use of

biclustering algorithms [33, 34]. The biclustering task aims to identify a set of biclusters $\mathcal{B} = (B_1, .., B_k)$, where each bicluster $B = (I, J)$ is an $n \times m$ subspace (subset of observations $I = \{i_1,.., i_n\} \subseteq X$ and subset of variables $J = \{j_1,.., j_m\} \subseteq Y$), that satisfy specific criteria:

- *homogeneity*—commonly guaranteed through the use of a merit function, such as the variance of the values in a bicluster [33], guiding the formation of biclusters in greedy, exhaustive, and stochastic/parametric searches determining their coherence, quality and structure;

- *statistical significance*—in addition to homogeneity criteria, guarantees that the probability of a bicluster's occurrence (against a null model) deviates from expectations [14];

- *dissimilarity*—criteria further placed to guarantee the absence of redundant biclusters (number, shape, and positioning) [35].

The bicluster **pattern** $\varphi_J$ is the set of expected values in the absence of adjustments and noise. A bicluster **pattern** is:

- constant overall if for all $i \in I$ and $j \in J$, $a_{ij} = \mu + \eta_{ij}$, where $\mu$ is the typical value and $\eta_{ij}$ is the observed noise;

- constant on columns, i.e. pattern on rows, if $a_{ij} = \mu_j + \eta_{ij}$, where $\mu_j$ represents the expected value in column $y_j$;

- additive if for all $i \in I$ and $j \in J$, $a_{ij} = \mu_j + \gamma_i + \eta_{ij}$ where $\mu_j$ represents the expected value in column $y_j$ and $\gamma_i$ the adjustment for observation $x_i$;

- multiplicative if for all $i \in I$ and $j \in J$, $a_{ij} = \mu_j \times \gamma_i + \eta_{ij}$ where $\mu_j$ represents the expected value in column $y_j$ and $\gamma_i$ the adjustment for observation $x_i$;

- order-preserving on variables if there is a permutation of $J$ under which the sequence of values in every row is strictly increasing. Likewise, order-preserving on observations if there is a permutation of $I$ under which the sequence of values in every columns is strictly increasing.

Fig 1 applies the aforementioned concepts.

The coverage $\Phi$ of the bicluster pattern $\varphi_J$, defined as $\Phi(\varphi_J)$, is the number of observations containing the bicluster pattern $\varphi_J$. The same logic can be applied to a nominal outcome of

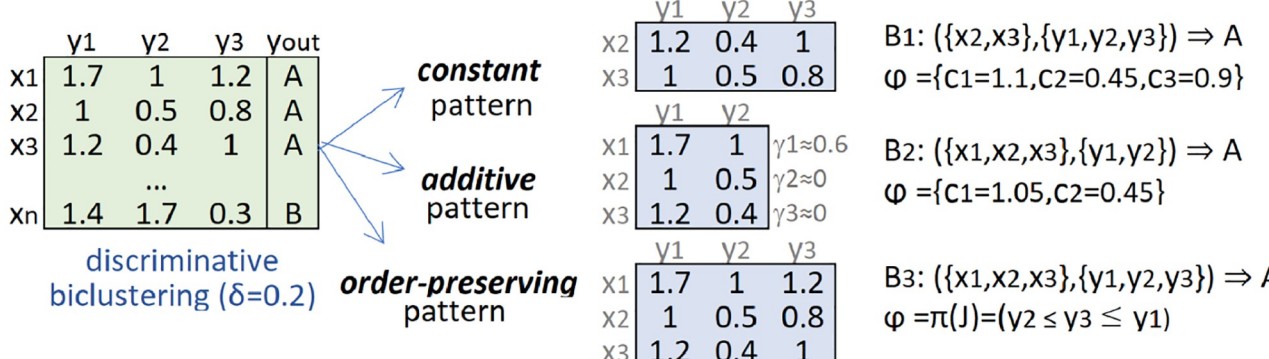

**Fig 1. Example of class-conditional subspaces with varying homogeneity.** The constant (on columns) subspace has pattern (value expectations) $\varphi_{J_1} = \{\mu_1 = 1.1, \mu_2 = 0.45, \mu_3 = 0.9\}$, the additive pattern $\varphi_{J_1} = \{\mu_1 = 1.1, \mu_2 = 0.45\}$ and $\{\gamma_1 = 0.6, \gamma_2 = 0, \gamma_3 = 0\}$, and the order-preserving subspace satisfies the $y_2 \leq y_3 \leq y_1$ permutation on 3 observations.

interest $c$, where $c$ can take any value in the class variable (e.g., $y_{out}$ in Fig 1). The coverage of the outcome, defined as $\Phi(c)$, is the number of observations with the outcome of interest.

Association rules describe a link between two events. An association rule is formed by two sides, the left-hand side (antecedent) and the right-hand side (consequent). In this case, an association rule can take the form $\varphi_J \to c$, where a pattern in the antecedent discriminates an outcome of interest in the consequent. The coverage of the association rule, $\Phi(\varphi_J \to c)$, is given by the number of observations where both the pattern $\varphi_J$ and the outcome $c$ co-occur.

Through the use of interestingness measures, an association rule can be assessed with respects to its' interestingness, statistical significance, usefulness, information gain, discriminative power, amongst others [15]. Two well-established interestingness measures are the *confidence*, $\Phi(\varphi_J \to c)/\Phi(\varphi_J)$, measuring the probability of $c$ occurring when $\varphi_J$ occurs, and, the *lift*, $(\Phi(\varphi_J \to c)/(\Phi(\varphi_J) \times \Phi(c)) \times N$, that further considers the probability of the consequent to assess the dependence between the consequent and antecedent.

A simple extension of the interestingness measures to accommodate continuous output variables is through the use of an numerical interval of interest. In the context of this extension, the coverage of the outcome $\Phi(c)$ can now be rewritten as $\Phi([v_1, v_2])$, where $v_1$ represents the lower bound of the interval and $v_2$ the upper bound, and the coverage of the association rule as $\Phi(\varphi_J \to [v_1, v_2])$. Understandably, the outcomes conditioned to a pattern of interest can be described by a probability density function (pdf). In this context, mapping the outcomes into a simple numerical range is generally inadequate as the pdf of pattern-conditional outcomes is often non-uniform and its discriminative properties can only be determined against the remaining observations.

## Methods

### Proposed approach

The proposed methodology allows for a robust analysis of the discriminative properties of a pattern in the presence of numerical outcome variables without imposing predefined rigid boundaries. Given a pattern $\varphi_J$, we first compare the underlying distributions of the outcome variable of interest, $z$, for the overall observations, $p(z|X)$, and the pattern coverage, $p(z|\Phi(\varphi_B))$, in order to extract numeric ranges, that compose the consequent, $\varphi_J \to \bigcup_i [v_1^{(i)}, v_2^{(i)}]$. Observations with the targeted pattern have higher likelihood to have numerical outcomes in the extracted range. Both empirical and theoretical distributions are allowed for this calculus. If instead of considering the underlying distributions to extract a range of values, we considered just the minimum and maximum values within the pattern, the likelihood of the target pattern having high values of discriminative properties would be lessened due to: 1) presence of outliers that make the interval more relaxed, and 2) the possibility of the interval being to rigid and excluding nearby values outside after and before the maximum and minimum values.

To illustrate these concepts, Fig 2 provides an example with two theoretical distributions approximated from the overall and pattern-conditional targets, respectively. By estimating the relative frequency of each of the distributions, two points of intersection $v_1$ and $v_2$ can be calculated, composing an interval that can be potentially discriminated by observations with the given pattern.

Once ranges of outcomes of interest are identified, classic interestingness measures for association rules can be extended to handle these consequents. Considering the previously introduced lift function—a paradigmatic function to assess the discriminative power of an

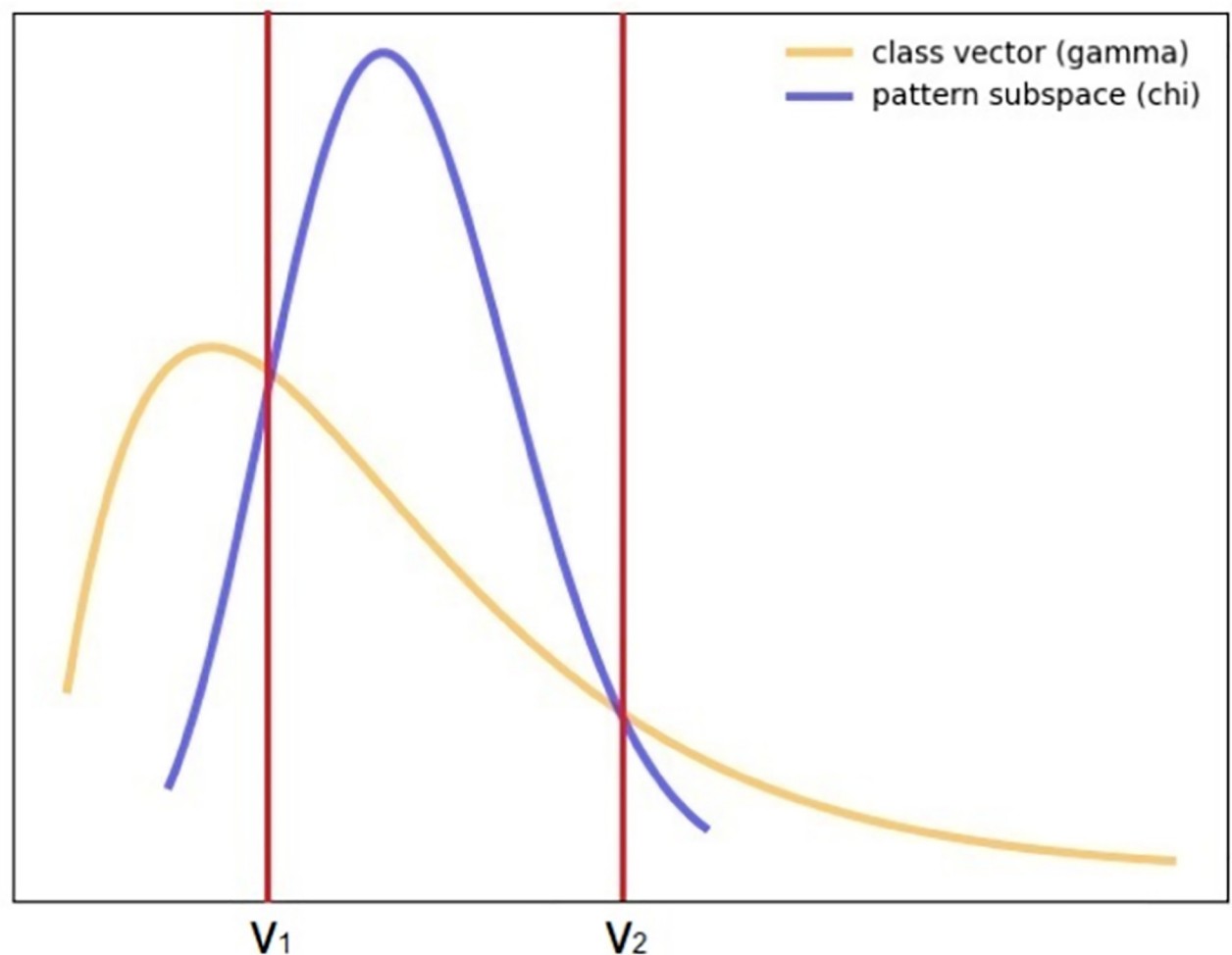

**Fig 2. Intersection of two theoretical distributions.** The yellow line represents the outcome variable, which follows a gamma distribution; The blue line represents the pattern-conditional outcome variable, which follows a $\chi^2$ distribution. In this example, the two points of intersection between the distributions form an interval, $[v_1, v_2]$. Observations with the targeted pattern have higher likelihood to have numerical outcomes in the extracted range.

association rule –, it can now be rewritten,

$$lift(\varphi_J \rightarrow [v_1, v_2]) = \frac{P(\varphi_J \cap [v_1, v_2])}{P(\varphi_J) \times P([v_1, v_2])} = \frac{\Phi(\varphi_J \rightarrow [v_1, v_2])}{\Phi(\varphi_J) \times \Phi([v_1, v_2])} \cdot \times N \qquad (1)$$

Note that the coverage of the outcome of interest is now defined as the interval created by the intersection points of the distributions. Instead of a predefined restrictive category range, intervals disclose outcomes of interest that are dynamically inferred for a given pattern in order to better assess its discriminative profile.

Consider now an example with two random empirical distributions originating more than two points of intersections, illustrated in Fig 3.

In this example, observations with a the selected pattern, have higher likelihood to show outcomes in the two inferred ranges. With two intervals, $[v_1, v_2]$ and $[v_3, v_4]$, we can further assess the discriminative power of the pattern with regards to each interval, as well as both

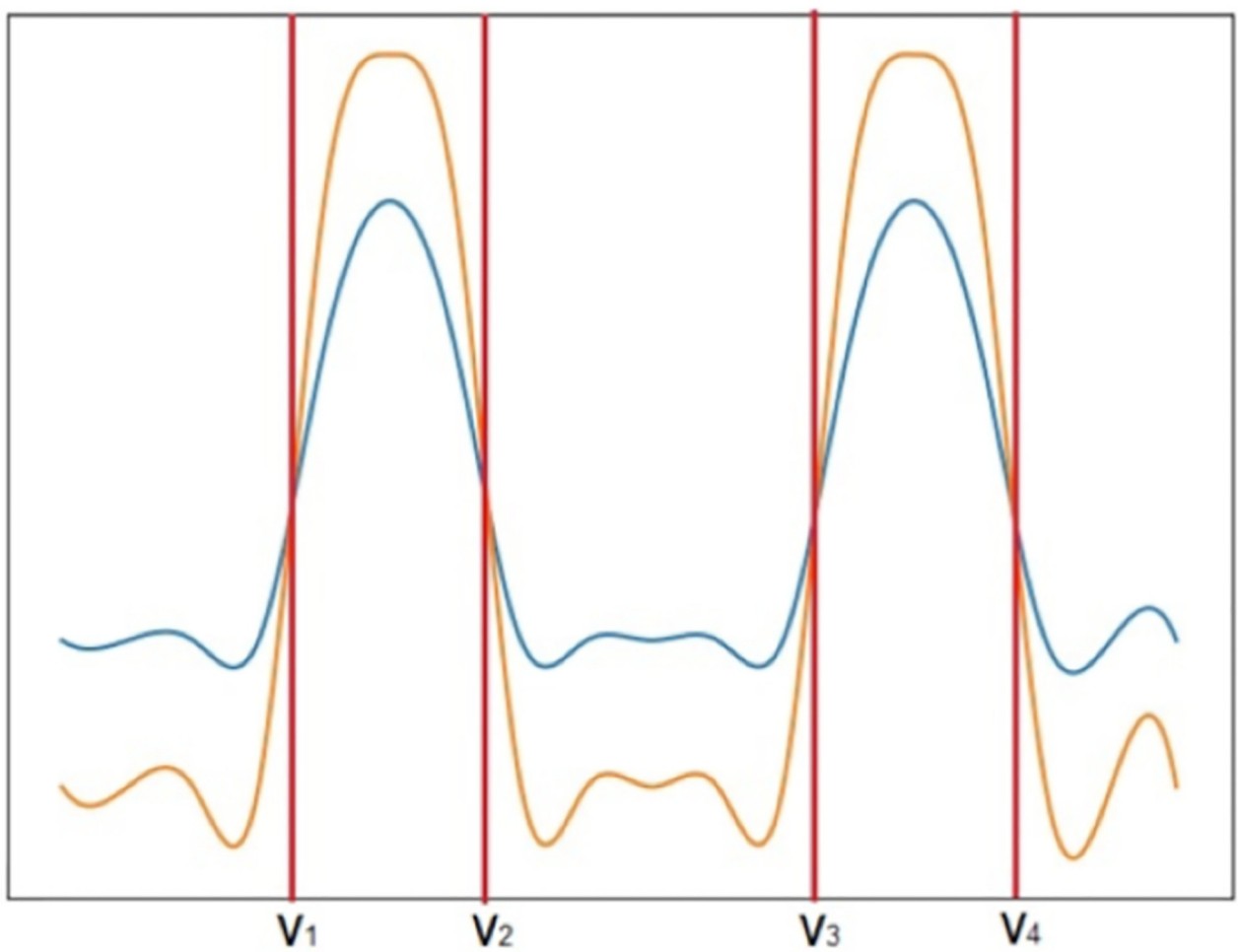

**Fig 3. Two empirical distributions represented with relative frequency bins.** The blue line represents the outcome variable, and the yellow line the pattern-conditional outcome variable. In this example, four points of intersection between the distributions form two intervals with ranges $[v_1, v_2]$ and $[v_3, v_4]$.

intervals,

$$lift_2(\varphi_J \to [v_1, v_2]) = \frac{\Phi(\varphi_J \to [v_1, v_2])}{\Phi(\varphi_J) \times \Phi([v_1, v_2])} \times N \tag{2}$$

$$lift_3(\varphi_J \to [v_3, v_4]) = \frac{\Phi(\varphi_J \to [v_3, v_4])}{\Phi(\varphi_J) \times \Phi([v_3, v_4])} \times N \tag{3}$$

$$lift_1(\varphi_J \to ([v_1, v_2] \cup [v_3, v_4])) = \frac{\Phi(\varphi_J \to ([v_1, v_2] \cup [v_3, v_4]))}{\Phi(\varphi_J) \times \Phi([v_1, v_2] \cup [v_3, v_4])} \times N \tag{4}$$

Different discriminative criteria can be considered in the presence of consequents given by multiple ranges. Considering the lift as the illustrative case, the discriminated outcomes can be

given by the numerical interval that maximises the lift function,

$$argmax_{c_i}\{lift(\varphi_J \rightarrow c_i)\}, \tag{5}$$

where, for the given example, $c_i \in \{([v_1, v_2] \cup [v_3, v_4]), [v_1, v_2], [v_3, v_4]\}$. All numerical intervals where lift satisfies a minimum threshold $\theta$,

$$\{c_i \mid lift(\varphi_J \rightarrow c_i). \geq \theta\}. \tag{6}$$

Both are valid options and allow for a robust analysis of the numerical outcome. The first approach retrieves the numerical interval, or combination of numerical intervals, with highest discriminative power. The second filters out uninformative/non-discriminative numerical intervals, allowing for a more comprehensive analysis of each pattern.

DISA implements the presented methodology and is given in Algorithm 1.

**Algorithm 1:** DISA tool

```
Input: data_matrix, class_vector, pattern_list, distribution
Output: list of statistics per pattern
statistics = [];
for p in pattern_list do
  if class_vector is continuous then
    if distribution == "empirical" then
      intervals = intersection(empirical_pdf(class_vector), empiri-
cal_pdf(p));
    end
    if distribution == "gaussian" then
      intervals = intersection(gaussian_pdf(class_vector), gaus-
sian_pdf(p));
    end
    if distribution == "min_max" then
      intervals = [p.min(), p.max()];
    end
    if distribution == "average" then
      m = p.mean();
      std = p.std();
      intervals = [m-std, m+std];
    end
    temp_class_vector = discretize(intervals);
    pattern_properties = properties(data_matrix, p,
temp_class_vector);
  else
    pattern_properties = properties(data_matrix, p, class_vector);
  end
  statistics.append(objective_functions(pattern_properties))
end
return statistics;
```

When analysing the subspace in the presence of a continuous output variable DISA implements four different setups: 1) `MinMax`, where the cut-off points, $[v_1, v_2]$, correspond to the minimum and maximum pattern-conditional outcomes, respectively; 2) `Average`, where $v_1$ and $v_2$ are the bounds formed by considering the standard deviation from the average, $\mu - \sigma$ and $\mu + \sigma$, respectively, where $\mu$ ($\sigma$) is the average (standard deviation) of the pattern-conditional outcomes; 3) `Gaussian`, where we assume that both the output variable and the pattern-conditional outcomes follow a normal distribution. In this case, $v_1$ and $v_2$ correspond to the intersection points between the gaussians where the range $[v_1, v_2]$ represents the most probable values of interest that the pattern discriminates. Fig 4 provides a in-depth example with three distinct patterns; 4) `Empirical`, where we assume they follow their own unique

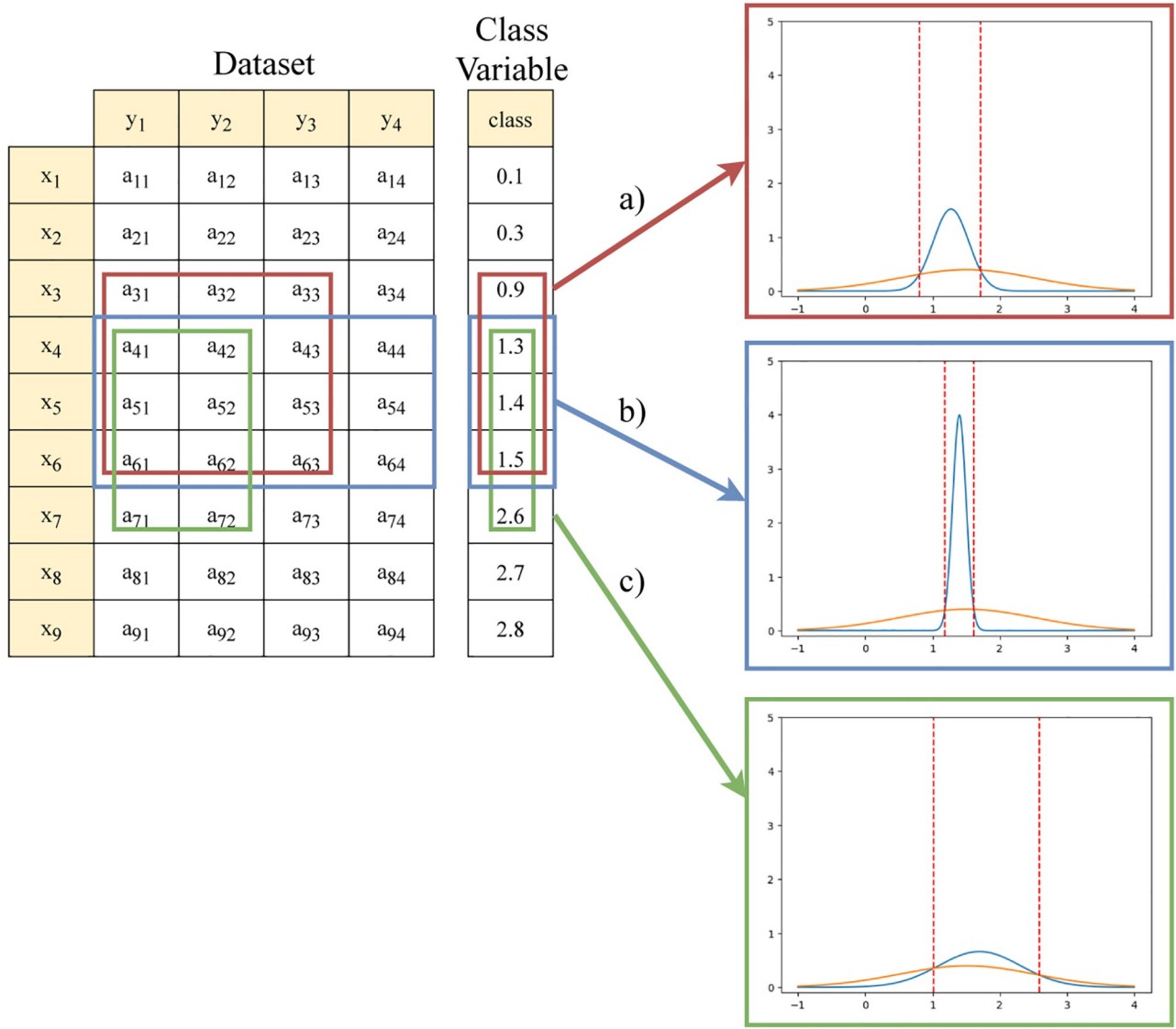

**Fig 4. An illustrative in-depth example.** Consider a dataset with observations $X=\{x_1,.., x_9\}$, variables $Y=\{y_1,.., y_4, class\}$, and a set of three association rules. By intercepting the output variable pdf with the pdfs of each rules' consequent, we obtain the following intervals: a) [0.80, 1.70], b) [1.18, 1.61], and c) [1.02, 2.59]. With this, discriminative power statistics, such as lift, can be computed. In this case lift is equal to: a) 2.27, b) 3.03, and c) 2.27.

empirical distribution, instead of assuming that both the outcome variable and the pattern-conditioned outcomes follow a well-known theoretical continuous distribution. In this case, $v_1$ and $v_2$ might not be the only points of intersection. We assume there can be any number between one and $n$ points of intersection. Fig 3 provides an example with four points of intersection, creating two ranges of intervals of interest. However, it is important to note that the number of intervals created is not necessarily correlated with the number of points of intersection. When the relative frequency of the pattern-outcome starts, or finishes, above the relative frequency of the output variable, the number of intervals changes. In Fig 5 we present three cases where the aforesaid happens.

To calculate the intersection points in linear time, $O(N)$, where $N$ represents the number of observations in the output variable, DISA executes the following steps: i) calculate the relative

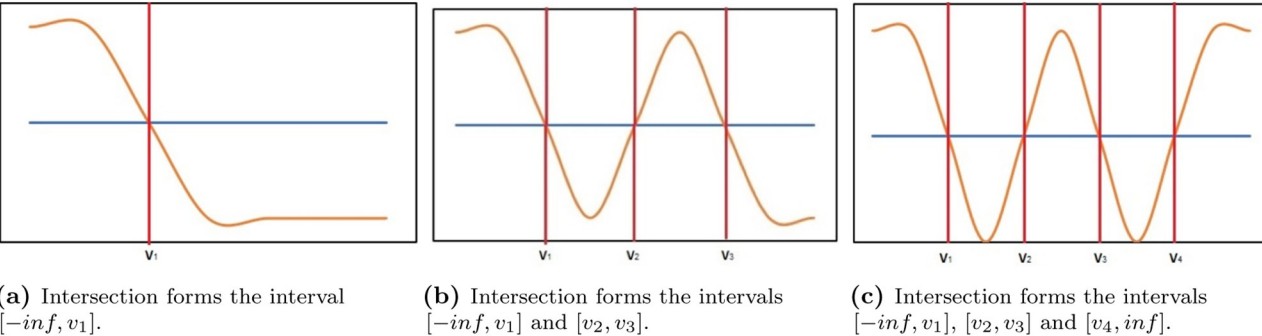

**(a)** Intersection forms the interval $[-inf, v_1]$.

**(b)** Intersection forms the intervals $[-inf, v_1]$ and $[v_2, v_3]$.

**(c)** Intersection forms the intervals $[-inf, v_1]$, $[v_2, v_3]$ and $[v_4, inf]$.

**Fig 5. Special cases of the intersection between empirical distributions.** The blue line represents the outcome variable, and the yellow line the pattern-conditional outcome variable. (a) Intersection forms the interval [−inf, v1]. (b) Intersection forms the intervals [−inf, v1] and [v2, v3]. (c) Intersection forms the intervals [−inf, v1], [v2, v3] and [v4, inf].

frequency of each unique value for both the overall and pattern-conditioned outcomes, ii) element-wise subtraction between the arrays, iii) extract the element-wise indication of the sign of each number on the resulting array, iv) calculate the discrete difference along the sign vector (value at position *i+1* minus value at position *i*), and finally v) find the indices of elements that are non-zero, grouped by element.

Consider a practical example where *outputs* = [1, 3, 4, 5, 7] and *pattern-conditioned outputs* = [3, 4, 5]. Accordingly, i) relative frequency conversion yields *outputs* = [0.2, 0.2, 0.2, 0.2, 0.2] and *pattern-conditioned outputs* = [0.0, 0.3(3), 0.3(3), 0.3(3), 0.0], ii) element-wise subtraction returns [0.2, −0.1(3), −0.1(3), −0.1(3), 0.2], iii) sign extraction returns [1, −1, −1, −1, 1], iv) differencing operation leads to [−2, 0, 0, 2], finally, v) the indices that are non-zero will produce the intersection points [$v_1 = 0$, $v_2 = 3$] that map to the original values of 1 and 5.

As previously mentioned, the intersection of empirical distributions can generate more than one interval of interest. By default DISA considers all the of the pattern-conditioned outcome intervals to compute the discriminative and informative properties of the pattern. However, if the discriminative power of the pattern is still below a minimum threshold (e.g., lift<1.3), then DISA will start to disregard uninformative intervals. Starting from the lowest individually ranked (e.g., by lift), the intervals are disregarded one by one, until either all of them are removed (resetting to the default behavior) or the satisfaction of the minimum discriminative power.

## Software

The previously introduced methodology is made available as an open-source software package, DISA, developed in Python (v3.7). DISA is able to assess the discriminative properties from the inputted patterns in the presence of numeric or categorical outcome variables. A pipeline of the DISA package is illustrated in Fig 6. If DISA receives a numerical outcome, the outcome ranges that are likely to be discriminated by the observations supporting a given pattern are first determined. DISA accomplishes this by approximating two probability density functions (e.g. Gaussians), one for all the observed targets and the other with targets of the pattern coverage. The intersecting points between the two probability density functions is computed to identify the range of values discriminated by the pattern. Second, DISA extends state-of-the-art statistics for assessing the informative and discriminative power of classic association rules. Currently, DISA supports 53 evaluation metrics in total. An illustrative subset of metrics is provided in Table 1 (complete list in DISA's GitHub repository).

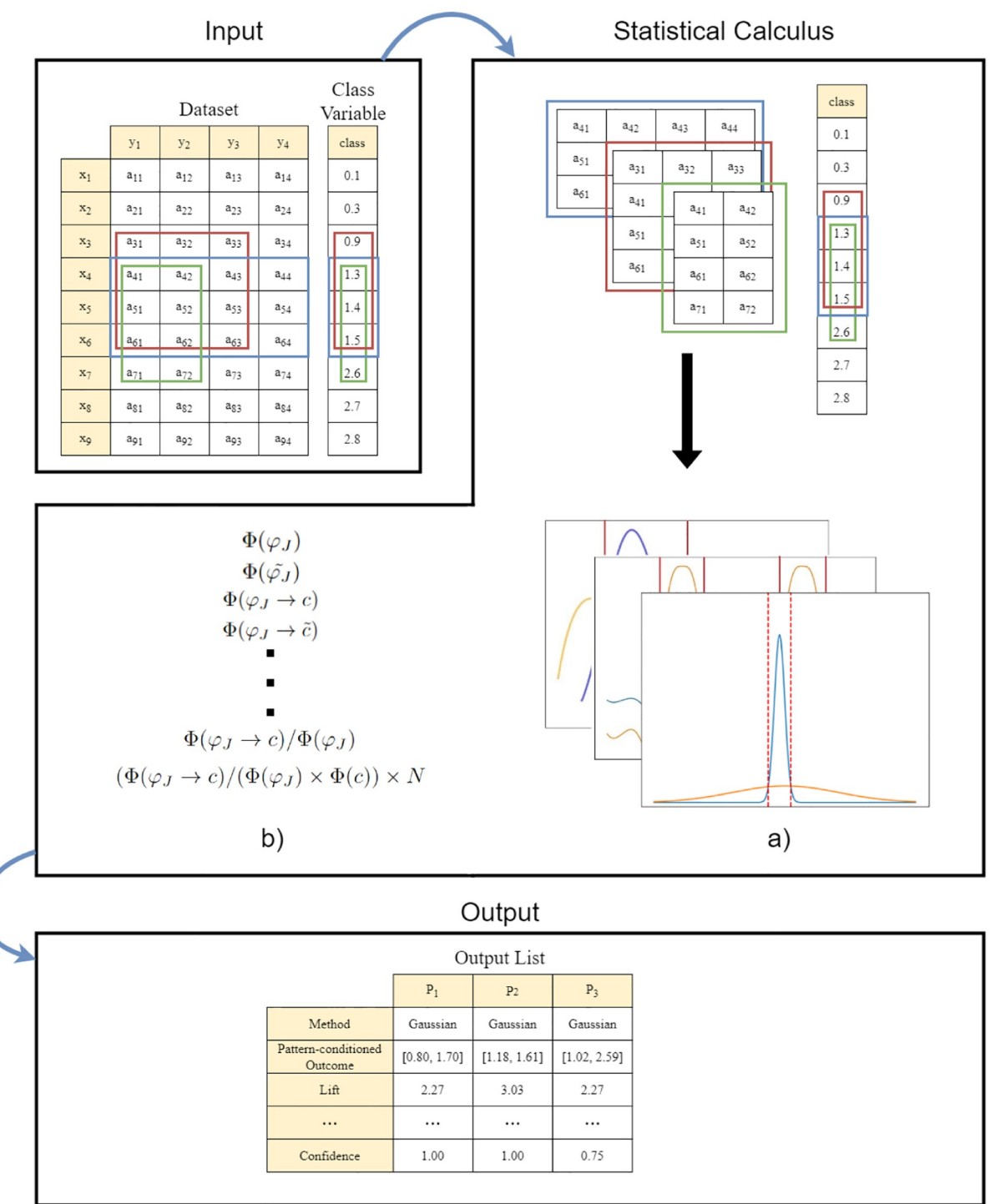

**Fig 6. Overview of DISA workflow.** Input: multivariate data (optional); list of patterns; and outcome variable. Statistical calculus: a) discriminated ranges from pdf (probability density function) intersection points (numerical outcomes only) and b) pattern properties, and metrics (e.g. statistical significance, gini index, information gain). Output: list of metrics per pattern.

**Table 1. A small sample of metrics implemented in DISA.** Three types are presented: support-based metrics, confidence-based, and lift-based.

| Support-based | Confidence-based | Lift-based |
|---|---|---|
| Statistical significance [14] | Confidence [36] | Lift [36] |
| Coverage (Support) [36] | All-Confidence [37] | Standardised Lift [16] |
| Difference in Support [11] | Casual Confidence [38] | Hyper Lift [39] |
| Bigger Support [11] | Descriptive Confirmed Confidence [40] | Weighted Lift [17] |
| Casual Support [38] | Hyper Confidence [39] | - |
| Weighted Support [17] | Laplace Corrected Confidence [40] | - |
| Weighted Rule Support [17] | Weighted Confidence [17] | - |

## Results

In order to illustrate DISA properties, we considered four public datasets taken from the literature: 1) *Echocardiogram* [41], monitoring physiological features of patients that suffered heart attacks at some point in time, the task consists in extracting discriminative patterns of survivability after a heart attack; 2) *Liver Disorders* [42], a dataset of molecular features from blood tests which are thought to mark liver disorders that might arise from excessive alcohol consumption, in this case the task consists in extracting patterns that discriminate the number of intake drinks per day; 3) *Breast Cancer Wisconsin (Diagnostic)* [43], where each observation corresponds to the follow-ups of a breast cancer patient, variables concern cancer cell nuclei features from a digitized image of a fine needle aspirate, and the outcome is the number of months until cancer relapse; and 4) *Dodecanol production* [20, 21], a dataset that monitors the concentration of key enzymes observed in the two Design-Build-Test-Learn cycles of 1-dodecanol production (a medium-chain fatty acid used in detergents, pharmaceuticals and cosmetics) in *Escherichia coli* with the outcome determining the concentration of the targeted organic compound. The list of variables per dataset, as well as their meaning, is presented in Table 2.

We used BicPAMS software [35] to extract patterns, with particular focus on constant coherence on columns (pattern on rows). Regarding statistical significance, we did not filter patterns exhibiting a p-value above 0.05 (patterns that might have occurred by chance). To allow the creation of larger patterns during the merging step, we allow up to 30% noise within the pattern. This will reduce the number of redundant patterns. Numeric input variables are categorised with DI2 discretizer [25], with $|L| = 3$, $|L| = 5$, and $|L| = 7$ categories.

Broad characteristics of the extracted patterns are presented in Tables 3 and 4. A set of illustrative patterns for each dataset are displayed in Fig 7, with the respective properties in Table 5.

## Discussion

In this work, we proposed an approach for pattern evaluation in the presence of numerical outcomes. Below, we experimentally assess the results of the proposed methodology on four publicly available datasets.

### Case study: *Echocardiogram*

A few discriminative patterns, yielding *Support* < 10%, were extracted from the *Echocardiogram* data as shown in Table 3. In Table 4, we can see that the most prominent discriminative criteria for the found patterns were an average *Lift* ≥ 1.6 and *StandardisedLift* ≥ 0.75 for configuration $|L| = 3$. A pattern with lift above 1 and *Standardised Lift* in [0.7,1] generally

**Table 2. Variable description for the four selected datasets for this study.** Variables not presented in this table were removed due to redundancy/irrelevance.

| Dataset name | Variables | Type |
|---|---|---|
| *Echocardiogram* (130 samples) | **age-at-heart-attack:** Age in years when heart attack occurred | Integer |
| | **pericardial-effusion:** Pericardial effusion is fluid around the heart. | Binary |
| | **fractional-shortening:** Contracility around the heart (lower numbers are increasingly abnormal) | Continuous |
| | **epss:** E-point septal separation, another measure of contractility | Continuous |
| | **lvdd:** Left ventricular end-diastolic dimension (size of the heart at end-diastole) | Continuous |
| | **wall-motion-index:** A measure of how the segments of the left ventricle are moving. | Continuous |
| | **survival** (*outcome*): Survivability in months | Integer |
| *Liver Disorders* (344 samples) | **mcv:** mean corpuscular volume | Continuous |
| | **alkphos:** alkaline phosphotase | Continuous |
| | **sgpt:** alanine aminotransferase | Continuous |
| | **sgot:** aspartate aminotransferase | Continuous |
| | **gammagt:** gamma-glutamyl transpeptidase | Continuous |
| | **drinks** (*outcome*): number of half-pint per day | Continuous |
| *Breast Cancer* Wisconsin (Diagnostic) (46 samples) | **radius:** mean of distance from center to points on the perimeter | Continuous |
| | **texture:** standard deviation of gray-scale values | Continuous |
| | **perimeter:** perimeter of the cell nucleus | Continuous |
| | **area:** area of the cell nucleus | Continuous |
| | **smoothness:** local variation in radius lengths | Continuous |
| | **compactness:** perimeter x perimeter / area−1 | Continuous |
| | **concavity:** severity of concave portions of the contour | Continuous |
| | **concave points:** number of concave portions of the contour | Continuous |
| | **symmetry:** symmetry of the cell nucleus enzyme that catalyzes oxidation-reduction (redox) reaction. | Continuous |
| | **fractal dimension:** coastline approximation | Continuous |
| | **tumor size:** diameter of the excised tumor in centimeters | Continuous |
| | **lymph node status:** number of positive axillary lymph node observed at time of surgery | Continuous |
| | **time** (*outcome*): Time of cancer recurrence in months | Continuous |
| *Dodecanol* (237 samples) | **AHR_ECOLI:** level of enzyme that catalyzes the reduction of a wide range of aldehydes | Continuous |
| | **LCFA_ECOLI:** level of enzyme that catalyzes the esterification of exogenous long-chain fatty acids into metabolically active CoA thioesters for subsequent degradation into phospholipids | Continuous |
| | **Dodecanoyl-[acyl-carrier-protein] hydrolase:** level of chloroplastic that plays an essential role in chain termination during de novo fatty acid synthesis | Continuous |
| | **Fatty acyl CoA reductase:** level of reduction catalyst of long chain acyl-CoA to fatty aldehyde | Continuous |
| | **A1U2T0:** level of enzyme that catalyzes long chain fatty acyl-CoA into a chain primary alcohol | Continuous |
| | **A1U3L3:** level of enzyme that catalyzes oxidation-reduction (redox) reaction. | Continuous |
| | **Dodecanol** (*outcome*): organic compound produced | Continuous |

discriminate the subspace of values it forms. The analysis of the found patterns using DISA reveals that the majority of discoveries discriminate a low survivability range (see GitHub repository https://github.com/JupitersMight/DISA/tree/main/Example for a detailed description of all patterns), including the pattern shown in Fig 7a. The patients suffering a heart attack in Fig 7a pattern exhibited moderate values of contractility and moderate size of the heart at end-diastole. In this case, when the local survivability within the pattern intersects the overall survivability, it forms a span of time that is discriminative of patients who survive a maximum of 12 months. This pattern possesses a high discriminative power, the maximum achievable, with a *Lift* = 3.09 and *StandardisedLift* = 1, and it also yields $\tilde{\chi}^2 = 8.64$. A $\tilde{\chi}^2 > 3.84$ means that the null-hypothesis of independence between the pattern and the outcome should be

**Table 3. Configurations used in BiCPAMS and best results from the four case studies.** Each row from left to right indicates the percentage of noise allowed, the number of categories for the continuous variables (coherence strength), the number of extracted patterns, the average number of columns in each patterns (and standard deviation), and the average number of rows in each pattern (and standard deviation).

| Configuration | | Echocardiogram | | | Liver Disorders | | | Breast Cancer Wisconsin | | | Dodecanol | | |
|---|---|---|---|---|---|---|---|---|---|---|---|---|---|
| Quality | $\|L\|$ | #bics | $\mu(\|I\|) \pm \sigma(\|I\|)$ | $\mu(\|J\|) \pm \sigma(\|J\|)$ | #bics | $\mu(\|I\|) \pm \sigma(\|I\|)$ | $\mu(\|J\|) \pm \sigma(\|J\|)$ | #bics | $\mu(\|I\|) \pm \sigma(\|I\|)$ | $\mu(\|J\|) \pm \sigma(\|J\|)$ | #bics | $\mu(\|I\|) \pm \sigma(\|I\|)$ | $\mu(\|J\|) \pm \sigma(\|J\|)$ |
| 100% | 3 | 3 | 9±4.3 | 3.3±1.2 | 7 | 42±19 | 2.5±0.7 | 56 | 9.8±1.3 | 2.3±0.8 | 6 | 36±16 | 3±0.8 |
| 100% | 5 | 8 | 7.1±2.9 | 2.8±0.7 | 8 | 20±10 | 2.2±0.4 | 50 | 6±1 | 2.5±1 | 18 | 18±6 | 2.1±0.3 |
| 100% | 7 | 13 | 5.6±1.2 | 2.5±0.4 | 16 | 8.7±4.1 | 2.4±0.4 | 96 | 3.5±0.8 | 2.8±1 | 25 | 11±8 | 2.7±0.6 |
| 70% | 3 | 3 | 9±4.3 | 3.3±1.2 | 7 | 42±19 | 2.5±0.7 | 59 | 10±1.4 | 2.3±0.7 | 7 | 29±13 | 3.1±0.8 |
| 70% | 5 | 8 | 7.6±3 | 2.8±0.7 | 8 | 20±10 | 2.2±0.4 | 54 | 6±1 | 2.4±1 | 20 | 17±7 | 2.4±0.8 |
| 70% | 7 | 14 | 5.7±1.4 | 2.6±0.6 | 16 | 8.7±4.1 | 2.4±0.4 | 102 | 3.6±0.9 | 2.9±1.4 | 24 | 12±8 | 2.6±0.6 |

rejected. Larger chi-squared values indicate stronger evidence of a strong relationship between the pattern and the outcome.

## Case study: *Liver disorders*

The extracted patterns from this data source display a *Support* $\geq$ 10% in configuration $|L| = 3$, as shown in Table 3. As the cardinality of input variables increases (higher $|L|$), the most salient discriminative criteria are $\tilde{\chi}^2 \geq 3.31$, *Lift* $\geq 1.7$, *StandardisedLift* $\geq 0.80$, and *Stat. Significance* $\leq 0.03$ for configuration $|L| = 7$. In this context, a statistical significance lower than 0.05 means that the pattern probability of occurrence deviates from expectations. The careful analysis of the patterns using DISA revealed that a good portion of the discovered patterns discriminate a low drink intake per day (see GitHub repository for a trace of all patterns). An example is shown in Fig 7b, whose individuals with very high values of alanine aminotransferase and gama-glutamyl transpeptidase, generally drank up to 2 drinks per day. This pattern possesses a high discriminative power, the maximum achievable, with a *Lift* = 2.04 and *StandardisedLift* = 1. It also has a strong dependence with the outcome with a $\tilde{\chi}^2 = 5.28$, and statistical significance (*p-value* = 0.008).

## Case study: *Breast cancer wisconsin (Diagnostic)*

A high number of patterns were extracted from this data source as shown in Table 3. The patterns in general were borderline discriminative, with the most notable discriminative criteria being the *Lift* $\geq 1.3$ and *Stat.Significance* $\leq 0.04$ for configuration $|L| = 7$. The careful analysis of the patterns using DISA revealed that a significant number of the found patters discriminated a low time until cancer recurrence (images in GitHub). An example is shown in Fig 7c, whose patients with short periors until cancer relapse show high heterogeneity cells characteristics, a very high number of compact cells and high severity of concave portions of the contour. In this case, the dynamically inferred discriminative span of time until relapse is between 1.4 and 9 months. The pattern possesses a high discriminative power, the maximum achievable, with a *Lift* = 4.18 and *StandardisedLift* = 1. It also has a $\tilde{\chi}^2 = 10.21$, meaning that the pattern is strongly dependent of the given span of time for cancer reoccurrence, and statistical significance (*p-value* = $8.63 \times 10^{-6}$).

## Case study: *Dodecanol*

Finally in the *Dodecanol* dataset, for configuration $|L| = 7$, a moderate number of patterns were extracted as shown in Table 3. The discriminative criteria is optimal across the found patterns from all configurations, e.g. $\tilde{\chi}^2 \geq 16$, *Lift* $\geq 1.5$, *StandardisedLift* $> 0.9$, and *Stat.*

**Table 4. Results of the analysis of DISA (gaussian) across all patterns per dataset.** A selective list of statistical measures is provided. For each measure, the average value obtained across the patterns per parameterization, as well as the standard deviation, are presented.

| Parameterization | | Echocardiogram | | | | | |
|---|---|---|---|---|---|---|---|
| Quality | $|L|$ | Information Gain | Gini Index | $\tilde{\chi}^2$ | Lift | Standardise Lift | Stat. Significance |
| 100% | 3 | 0.06±0.05 | 0.01±0 | 2.69±0.88 | 1.64±0.26 | 0.75±0.17 | 0.05±0.03 |
| 100% | 5 | 0.03±0.02 | 0±0 | 1.57±1.17 | 0.96±0.45 | 0.43±0.21 | 0.07±0.06 |
| 100% | 7 | 0.04±0.06 | 0±0 | 1.71±2.18 | 1.01±0.66 | 0.41±0.19 | 0.06±0.05 |
| 70% | 3 | 0.06±0.05 | 0.01±0 | 2.69±0.88 | 1.64±0.26 | 0.75±0.17 | 0.05±0.03 |
| 70% | 5 | 0.02±0.01 | 0±0 | 1.44±0.96 | 0.92±0.38 | 0.41±0.19 | 0.07±0.06 |
| 70% | 7 | 0.04±0.06 | 0±0 | 1.88±2.08 | 1.09±0.67 | 0.44±0.21 | 0.07±0.05 |
| Parameterization | | Liver Disorders | | | | | |
| Quality | $|L|$ | Information Gain | Gini Index | $\tilde{\chi}^2$ | Lift | Standardise Lift | Stat. Significance |
| 100% | 3 | 0.01±0.01 | 0±0 | 2.72±3.56 | 1.10±0.08 | 0.78±0.05 | 0.09±0.10 |
| 100% | 5 | 0.01±0.01 | 0±0 | 2.36±1.61 | 1.18±0.28 | 0.69±0.17 | 0.05±0.03 |
| 100% | 7 | 0.06±0.07 | 0±0 | 3.31±2.98 | 1.7±0.75 | 0.80±0.13 | 0.03±0.03 |
| 70% | 3 | 0.01±0.01 | 0±0 | 2.72±3.56 | 1.10±0.08 | 0.78±0.05 | 0.09±0.10 |
| 70% | 5 | 0.1±0.01 | 0±0 | 2.36±1.61 | 1.18±0.28 | 0.69±0.17 | 0.05±0.03 |
| 70% | 7 | 0.06±0.07 | 0±0 | 3.31±2.98 | 1.7±0.75 | 0.80±0.13 | 0.03±0.03 |
| Parameterization | | Breast Cancer Wisconsin (Diagnostic) | | | | | |
| Quality | $|L|$ | Information Gain | Gini Index | $\tilde{\chi}^2$ | Lift | Standardise Lift | Stat. Significance |
| 100% | 3 | 0.05±0.05 | 0.2±0.2 | 2.12±2.16 | 1.17±0.28 | 0.71±0.21 | 0.08±0.07 |
| 100% | 5 | 0.03±0.04 | 0.01±0.01 | 1.11±1.29 | 1.19±0.26 | 0.69±0.17 | 0.03±0.04 |
| 100% | 7 | 0.07±0.10 | 0.01±0.02 | 1.72±2.65 | 1.25±0.81 | 0.61±0.27 | 0.05±0.06 |
| 70% | 3 | 0.05±0.07 | 0.02±0.02 | 2.18±2.49 | 1.2±0.27 | 0.72±0.19 | 0.08±0.07 |
| 70% | 5 | 0.05±0.07 | 0.01±0.01 | 1.49±1.88 | 1.25±0.35 | 0.72±0.16 | 0.04±0.04 |
| 70% | 7 | 0.07±0.11 | 0.01±0.02 | 1.87±3.01 | 1.3±0.92 | 0.62±0.27 | 0.04±0.05 |
| Parameterization | | Dodecanol | | | | | |
| Quality | $|L|$ | Information Gain | Gini Index | $\tilde{\chi}^2$ | Lift | Standardise Lift | Stat. Significance |
| 100% | 3 | 0.22±0.13 | 0.06±0.03 | 34.44±20.35 | 1.94±0.74 | 0.92±0.12 | 0.03±0.06 |
| 100% | 5 | 0.17±0.13 | 0.03±0.03 | 20.05±17.14 | 2.08±0.91 | 0.9±0.11 | 0.04±0.04 |
| 100% | 7 | 0.19±0.11 | 0.03±0.02 | 16.71±14.45 | 2.51±1 | 0.92±0.13 | 0.03±0.04 |
| 70% | 3 | 0.1±0.1 | 0.03±0.04 | 19.21±21.55 | 1.49±0.5 | 0.87±0.10 | 0.06±0.06 |
| 70% | 5 | 0.16±0.13 | 0.03±0.03 | 18.34±17.07 | 2.05±0.89 | 0.9±0.12 | 0.03±0.04 |
| 70% | 7 | 0.19±0.12 | 0.03±0.02 | 17.18±14.76 | 2.45±0.96 | 0.92±0.13 | 0.03±0.04 |

*Significance* $\leq 0.04$. The careful analysis of the patterns using DISA revealed that some of the discovered patterns discriminated a low production of dodecanol (images in GitHub). An example is shown in Fig 7d, where a reduced dodecanol production (maximum of 0.14 units) is discriminated by the presence of samples with a very high concentration of enzymes responsible for the catalysis of long chain fatty acyl-CoA into a chain primary alcohol and low concentration of enzymes responsible for oxidation-reduction (redox) reactions. The pattern possesses moderate discriminative power with a *Lift* = 1.49 and *StandardisedLift* = 0.87. It further shows a strong relation with the outcome, $\tilde{\chi}^2 = 12.8$, and possesses high statistical significance (*p-value* = $7.23 \times 10^{-6}$).

## State-of-the-art comparison

To test the DISA assessment of the patterns' discriminative power, we considered an additional set of approaches: 1) `Classic` approach, where the numerical outcome variable is

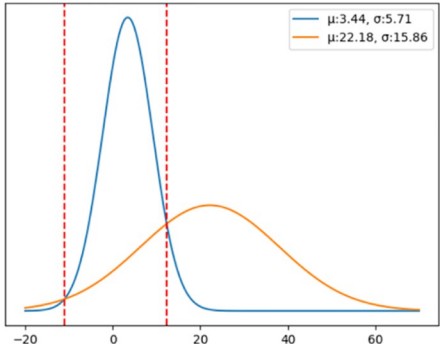

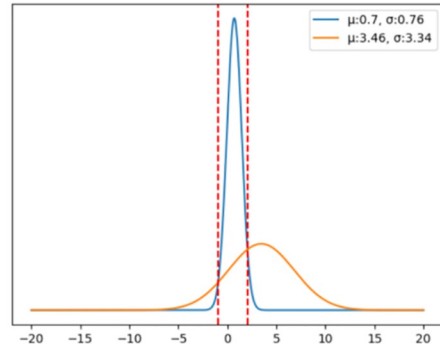

**(a)** *Echocardiogram* pattern with intersections at −11.04 and 12.33.

**(b)** *Liver Disorders* pattern with intersections at −0.94 and 2.04.

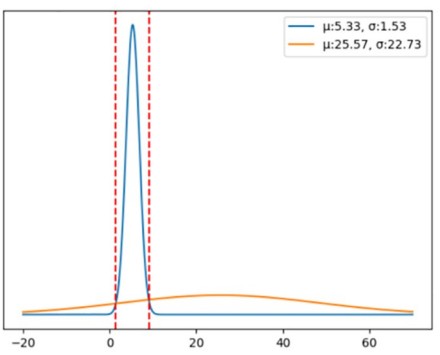

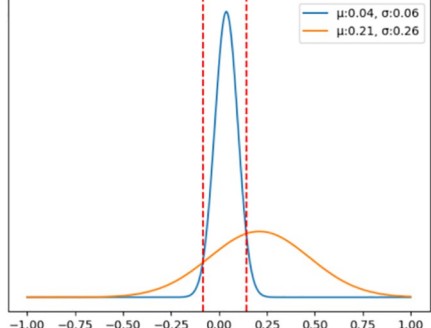

**(c)** *Breast Cancer Wisconsin (Diagnostic)* pattern with intersections at 1.43 and 9.05.

**(d)** *Dodecanol* pattern with intersections at −0.07 and 0.14.

**Fig 7. Visual characterization of the discriminated range of outcomes per pattern.** Each chart displays the Gaussian intersections between the outcome variable and the distribution of pattern-conditional outcomes. The blue line represents the Gaussian of the pattern outcome space, orange line represents the Gaussian of the original outcome space. All patterns displayed above have $|L| = 7$ categories (very low, low, medium-low, medium, medium-high, high, very high). Patterns: (a) $\varphi_{J_A}$ = {medium contractility (epss), medium size of the heart at end-diastole (lvdd)}; (b) $\varphi_{J_B}$ = {medium-high values of alkaline phosphotase (alkphos), very-high values of alanine aminotransferase (sgpt), very-high values of gamma-glutamyl transpeptidase (gammagt)}; (c) $\varphi_{J_C}$ = {high standard deviation among the cells compactness, very high compactness, very high severity of concave portions of the contour, very high symmetry}; (d) $\varphi_{J_D}$ = {very high values of A1U2T0, low values of A1U3L3}. (a) Echocardiogram pattern with intersections at −11.04 and 12.33. (b) Liver Disorders pattern with intersections at −0.94 and 2.04. (c) Breast Cancer Wisconsin (Diagnostic) pattern with intersections at 1.43 and 9.05. (d) Dodecanol pattern with intersections at −0.07 and 0.14.

discretized by applying DI2 [25] with $|L| = 7$, and the outcome is then interpreted as a class. In this case, DISA selects for each pattern the best fitting class ordered by lift. It is important to note that the outcome class is defined prior to the discovery of the pattern, without assumptions related with the subsequently mined pattern-conditioned outcomes; 2) MinMax approach, that uses the minimum and maximum of the pattern-conditional outcomes, 3) Average approach, that uses the average of the pattern-conditioned outcome with bounds inferred using the observed standard deviation; and 4) Empirical approach, where DISA considers the empirical distributions of both the continuous outcome variable and the pattern-conditioned outcome variable. These novel approaches are applied to each pattern illustrated in Fig 7. Table 5 contains the results of this analysis and we will discuss the results of: i) the proposed Gaussian approach versus Classic and standard approaches; and ii) the proposed Empirical approach versus all others.

**Table 5. Value of objective interestingness functions for each pattern in Fig 7.** In this analysis, we compare the discriminative assessment produced over the range of discriminated outcomes with DISA against classic alternatives produced by discretizing the numerical outcome using DI2 [25] and the standard `MinMax` and `Average` approach (see Methods section) for the four datasets. The reference DISA values per function are presented in the last two rows.

| | | Information Gain | Gini Index | $\tilde{\chi}^2$ | Lift | Standardised Lift | Stat. Significance |
|---|---|---|---|---|---|---|---|
| *Echocardiogram* | | | | | | | |
| DI2 | Classic [0.03, 0.59] | 0.17 | 0.02 | **8.76** | **4.06** | **0.66** | 0.16 |
| Standard approach | MinMax [0.5, 12.0] | 0.29 | 0.03 | **10.37** | **3.51** | **1.00** | 0.16 |
| | average (±σ) [−2.27, 9.14] | 0.12 | 0.02 | **5.10** | **2.87** | **0.67** | 0.16 |
| DISA | Gaussian [−11.04, 12.33] | 0.26 | 0.02 | **8.64** | **3.09** | **1.00** | 0.16 |
| | Empirical [[0.25, 0.75], [11.0, 12.0]] | 0.43 | 0.04 | **21.42** | **6.19** | **1.00** | 0.16 |
| *Liver Disorders* | | | | | | | |
| | | Information Gain | Gini Index | $\tilde{\chi}^2$ | Lift | Standardised Lift | Stat. Significance |
| DI2 | Classic [0, 0.8] | 0.08 | 0.00 | **4.80** | **2.35** | **0.75** | **0.008** |
| Standard approach | MinMax [0.0, 2.0] | 0.14 | 0.01 | **5.28** | **2.04** | **1.00** | **0.008** |
| | average (±σ) [−0.05, 1.45] | 0.07 | 0.00 | 3.62 | **2.06** | **0.75** | **0.008** |
| DISA | Gaussian [−0.94, 2.04] | 0.13 | 0.007 | 5.28 | 2.04 | 1.00 | 0.008 |
| | Empirical [[0.0, 0.5], [1.0, 2.0]] | 0.14 | 0.01 | **5.28** | **2.04** | **1.00** | **0.008** |
| *Breast Cancer Wisconsin (Diagnostic)* | | | | | | | |
| | | Information Gain | Gini Index | $\tilde{\chi}^2$ | Lift | Standardised Lift | Stat. Significance |
| DI2 | Classic [−2.08, 4.85] | 0.23 | 0.04 | **8.13** | **5.11** | 0.49 | **8.63×10⁻⁶** |
| Standard approach | MinMax [4.0, 7.0] | **0.7** | 0.11 | **26.32** | **9.2** | **1.00** | **8.63×10⁻⁶** |
| | average (±σ) [3.80, 6.86] | 0.40 | 0.05 | **19.40** | **10.22** | 0.50 | **8.63×10⁻⁶** |
| DISA | Gaussian [1.43, 9.05] | 0.41 | 0.08 | **10.21** | **4.18** | **1.00** | **8.63×10⁻⁶** |
| | Empirical [2.0, 7.0] | 0.62 | 0.11 | **21.4** | **7.67** | **1.00** | **8.63×10⁻⁶** |
| *Dodecanol* | | | | | | | |
| | | Information Gain | Gini Index | $\tilde{\chi}^2$ | Lift | Standardised Lift | Stat. Significance |
| DI2 | Classic [0, 0.002] | 0.04 | 0.01 | **9.67** | **1.89** | 0.45 | **7.23×10⁻⁶** |
| Standard approach | MinMax [0.0, 0.15] | 0.18 | 0.04 | **22.08** | **1.6** | **1.0** | **7.23×10⁻⁶** |
| | average (±σ) [−0.01, 0.10] | 0.05 | 0.02 | **9.72** | **1.49** | **0.77** | **7.23×10⁻⁶** |
| DISA | Gaussian [−0.07, 0.14] | 0.07 | 0.02 | **12.8** | **1.49** | **0.87** | **7.23×10⁻⁶** |
| | Empirical* | 0.39 | 0.12 | **63.02** | **2.7** | **1.0** | **7.23×10⁻⁶** |
| | | Information Gain | Gini Index | $\tilde{\chi}^2$ | Lift | Standardised Lift | Stat. Significance |
| | Metric interval | [0, 1] | [0, 1] | [0, inf] | [0, inf] | [0, 1] | [0, 1] |
| | Reference threshold | >0.6 | >0.6 | >3.84 | >1.3 | >0.6 | <0.05 |

*[[0.000, 0.00538], [0.00717, 0.00735], [0.03961, 0.04472], [0.05002, 0.05014], [0.05149, 0.05544], [0.05641, 0.06605], [0.06607, 0.07021], [0.10282, 0.11077], [0.13608, 0.13764], [0.1477, 0.14858], [0.15048, 0.15287], [0.15548, 0.15592]]

When comparing the results of the `Classic` and `Gaussian` approaches in the *Liver Disorders*, *Breast Cancer*, and *Dodecanol* datasets we observe that the `Gaussian` approach exhibits higher values in the $\tilde{\chi}^2$ function, whilst the `Classic` displays slight improvements in the *lift* function for most patterns. In spite of these results, the `Classic` approach fails to maximize the patterns' potential to discriminate specific ranges of outcomes. This can be concluded by observing a considerable decrease in the values of *Standardised Lift*. In the case of *Breast Cancer* and *Dodecanol*, *Standardised Lift* plummeted below 0.5 using the classic approach. The `Average` approach also exhibits this failure to find intervals that maximize the patterns' discriminative potential. These values confirm our initial hypothesis, that the classic and `Average` approaches form intervals that might not fully explore the discriminative profile of each pattern. The `MinMax` approach is able to fully accommodate noise and outlier pattern-

conditioned outcomes, yet it creates intervals that can be too large or permissive, i.e. intervals that accommodate outcome ranges that are not discriminated by the pattern. The `Gaussian` approach is in theory more robust to this problem, an observation that is corroborated by the collected results.

Considering the selected *Echocardiogram* pattern, two intervals are formed by the `Empirical` approach. The first interval captures a low survivability range, [0.25, 0.75] whilst the second captures a higher survivability range [11.0, 12.0]. If we observe the statistics of the other approaches that enclose either one or two of these intervals we can conclude: 1) that the interval of low survivability provides discriminative properties, i.e. in the `classic` approach the range [0.03, 0.59] partially encloses [0.25, 0.75]; and 2) that the approaches which consider the inclusion of higher values of survivability also display discriminative properties, i.e. a compact range that encloses both of the aforementioned intervals are observed in the `MinMax` and `Gaussian` approaches, [0.5, 12.0] and [−11.04, 12.33], respectively. Note, nevertheless that the `Empirical` approach disregards the range of values between [0.75, 11.0]. Results from all datasets confirm an increase in the discriminative potential in most statistics for all patterns (e.g., Standardised Lift as 1). However, the `Empirical` approach is generally restrictive in the formation of pattern-conditioned outcome intervals, and should be applied with care, e.g., complemented with the `Gaussian` approach to guarantee that the consequent of the target association rules include all numerical ranges discriminated by a given pattern.

## Conclusion

This work proposed a novel distribution-based method to rigorously assess association rules in the presence of numerical outcomes in the consequent, by inspecting the differences between the distribution of the numerical outcomes for all observations and those supporting a given pattern. This methodology allows for a dynamic and pattern-tailored approach to numerical outcomes as the patterns dictate how the discriminated ranges of values are statistically produced. The results further confirm the utility of the proposed methodology in dynamically producing pattern-tailored intervals, where discovered patterns from multiple domains exhibited maximum achievable discriminative power properties.

The methodology is implemented in DISA, an open source Python package capable of robustly assessing the statistical significance and discriminative power of association rules in the presence of numerical and categorical outcomes. DISA implements over 50 metrics, heuristics that can be used to guide the discovery process of discriminative patterns and subspace clusters in various data domains.We believe that DISA can be easily embed and further extended for more complex patterns, such as with multiple points of intersection, therefore aiding the scientific community ability along pattern-centric descriptive and predictive tasks. For instance, the extraction of patterns in omic data able to discriminate numerical phenotypes, or the extraction of patterns in clinical data able to discriminate risk scales.

## Author Contributions

**Conceptualization:** Leonardo Alexandre, Rui Henriques.

**Supervision:** Rafael S. Costa, Rui Henriques.

**Writing – original draft:** Leonardo Alexandre.

**Writing – review & editing:** Leonardo Alexandre, Rafael S. Costa, Rui Henriques.

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
