## [Decision Letter · Decision Letter 0]

1 Aug 2022

PONE-D-22-07846DISA tool: discriminative and informative subspace assessment with categorical and numerical outcomesPLOS ONE

Dear Dr. Alexandre,

Thank you for submitting your manuscript to PLOS ONE. After careful consideration, we feel that it has merit but does not fully meet PLOS ONE’s publication criteria as it currently stands. Therefore, we invite you to submit a revised version of the manuscript that addresses the points raised during the review process.

We look forward to receiving your revised manuscript.

Kind regards,

Sathishkumar V E

Academic Editor

PLOS ONE

https://journals.plos.org/plosone/s/file?id=ba62/PLOSOne_formatting_sample_title_authors_affiliations.pdf".

“This work was supported by the Associate Laboratory for Green Chemistry (LAQV), financed by national funds from FCT/MCTES (UIDB/50006/2020 and UIDP/50006/2020), INESC-ID plurianual (UIDB/50021/2020), the contract CEECIND/01399/2017 to RSC and the FCT individual PhD grant to LA (2021.07759.BD). This work was further supported by IPOscore with reference (DSAIPA/DS/0042/2018) and ILU (DSAIPA/DS/0111/2018).”

“This work was supported by the Associate Laboratory for Green Chemistry (LAQV), financed 269 by national funds from FCT/MCTES (UIDB/50006/2020 and UIDP/50006/2020), INESC-ID 270 plurianual (UIDB/50021/2020), the contract CEECIND/01399/2017 to RSC and the FCT 271 individual PhD grant to LA (2021.07759.BD). This work was further supported by IPOscore 272 with reference (DSAIPA/DS/0042/2018) and ILU (DSAIPA/DS/0111/2018).”

“This work was supported by the Associate Laboratory for Green Chemistry (LAQV), financed by national funds from FCT/MCTES (UIDB/50006/2020 and UIDP/50006/2020), INESC-ID plurianual (UIDB/50021/2020), the contract CEECIND/01399/2017 to RSC and the FCT individual PhD grant to LA (2021.07759.BD). This work was further supported by IPOscore with reference (DSAIPA/DS/0042/2018) and ILU (DSAIPA/DS/0111/2018).”

Reviewers' comments:

Reviewer's Responses to Questions

**Comments to the Author**

1. Is the manuscript technically sound, and do the data support the conclusions?

Reviewer #1: Yes

Reviewer #2: No

2. Has the statistical analysis been performed appropriately and rigorously? 

Reviewer #1: Yes

Reviewer #2: Yes

3. Have the authors made all data underlying the findings in their manuscript fully available?

Reviewer #1: Yes

Reviewer #2: Yes

4. Is the manuscript presented in an intelligible fashion and written in standard English?

Reviewer #1: Yes

Reviewer #2: No

5. Review Comments to the Author

Reviewer #1: The authors has produced various analysis to show his justification on the proposed mechanism,

1. why the authors say, "e discriminative properties 88

of a pattern in the presence of numerical outcome variables without imposing predefined

rigid boundaries."

2. For how many instances / or entities the association rule has been used or measured?

3. How DISA varies or limits from other approaches? why state of art is simplified?

4. Does the background parameters (homogeneity and others) taken in the analysis section to compare?

5. Strengthen the related study with

-Guo, Z., Yu, K., Jolfaei, A., Ding, F. and Zhang, N., 2021. Fuz-spam: label smoothing-based fuzzy detection of spammers in internet of things. IEEE Transactions on Fuzzy Systems.-

-Cheng, X., Guo, Z., Shen, Y., Yu, K. and Gao, X., 2021. Knowledge and data-driven hybrid system for modeling fuzzy wastewater treatment process. Neural Computing and Applications, pp.1-22.

-Saranya, A., Kottursamy, K., AlZubi, A.A. and Bashir, A.K., 2021. Analyzing fibrous tissue pattern in fibrous dysplasia bone images using deep R-CNN networks for segmentation. Soft Computing, pp.1-15.

Reviewer #2: The authors present a new method of clustering, in order to categories numerical outcomes; they illustrate their methods within biological application.

I think the work is not acceptable for publication; it is difficult to make an opinion about it for two reasons:

A) The presentation of the background and the method is not clear enough for the non-specialist: in the background part, some notions have not an explicit mathematical definition (“bi-cluster pattern”, “consequent”); the steps of the method are not presented. I suggest the author to reconsider “background” and “method” with exact and explicit mathematical definition of all object, and with a presentation of the method as a set of steps.

B) Although the illustrations of the method on real data seem promising, it is not clear weather this method performs better than other ones. I suggest the author to produce examples where their method is compared to other ones; in addition, it would be useful to know in which case their method is expected to perform better than other ones.

6. PLOS authors have the option to publish the peer review history of their article (what does this mean?). If published, this will include your full peer review and any attached files.

Reviewer #1: No

Reviewer #2: No

---

## [Author Response · Author response to Decision Letter 0]

17 Sep 2022

Dear Reviewers,

Thank you for all your attention and feedback.

We provide a letter containing the details on how the raised concerns were incorporated into the revised version of the manuscript.

The manuscript was carefully proof-checked, revised, and expanded to address the received comments.

The undertaken changes in the revised version are highlighted in blue.

---

## [Decision Letter · Decision Letter 1]

4 Oct 2022

DISA tool: discriminative and informative subspace assessment with categorical and numerical outcomes

PONE-D-22-07846R1

Dear Dr. Alexandre,

We’re pleased to inform you that your manuscript has been judged scientifically suitable for publication and will be formally accepted for publication once it meets all outstanding technical requirements.

Kind regards,

Sathishkumar V E

Academic Editor

PLOS ONE

Additional Editor Comments (optional):

Reviewers' comments:

Reviewer's Responses to Questions

**Comments to the Author**

1. If the authors have adequately addressed your comments raised in a previous round of review and you feel that this manuscript is now acceptable for publication, you may indicate that here to bypass the “Comments to the Author” section, enter your conflict of interest statement in the “Confidential to Editor” section, and submit your "Accept" recommendation.

Reviewer #1: All comments have been addressed

2. Is the manuscript technically sound, and do the data support the conclusions?

Reviewer #1: Yes

3. Has the statistical analysis been performed appropriately and rigorously? 

Reviewer #1: Yes

4. Have the authors made all data underlying the findings in their manuscript fully available?

Reviewer #1: Yes

5. Is the manuscript presented in an intelligible fashion and written in standard English?

Reviewer #1: Yes

6. Review Comments to the Author

Reviewer #1: The authors has addressed all the comments, i accept the paper

7. PLOS authors have the option to publish the peer review history of their article (what does this mean?). If published, this will include your full peer review and any attached files.

Reviewer #1: No

---

## [Editor Report · Acceptance letter]

10 Oct 2022

PONE-D-22-07846R1 

DISA tool: discriminative and informative subspace assessment with categorical and numerical outcomes 

Dear Dr. Alexandre:

I'm pleased to inform you that your manuscript has been deemed suitable for publication in PLOS ONE. Congratulations! Your manuscript is now with our production department. 

Kind regards, 

on behalf of

Dr. Sathishkumar V E 

Academic Editor

PLOS ONE